

# Analyzing moisture-heat coupling in a wheat-soil system using data-driven vector autoregression model

Xiaohang Feng[1,2], Xia Zhang[1], Zhenqi Feng[1] and Yichang Wei[1]

[1] North China University of Water Resources Electric Power, Zhengzhou, Henan, China
[2] Peking University, Beijing, China

## ABSTRACT

Soil temperature and moisture have a close relationship, the accurate controlling of which is important for crop growth. Mechanistic models built by previous studies need exhaustive parameters and seldom consider time stochasticity and lagging effect. To circumvent these problems, this study designed a data-driven stochastic model analyzing soil moisture-heat coupling. Firstly, three vector autoregression models are built using hourly data on soil moisture and temperature at the depth of 10, 30, and 90 cm. Secondly, from impulse response functions, the time lag and intensity of two variables' response to one unit of positive shock can be obtained, which describe the time length and strength at which temperature and moisture affect each other, indicating the degree of coupling. Thirdly, Granger causality tests unfold whether one variable's past value helps predict the other's future value. Analyzing data obtained from Shangqiu Experiment Station in Central China, we obtained three conclusions. Firstly, moisture's response time lag is 25, 50, and 120 h, while temperature's response time lag is 50, 120, and 120 h at 10, 30, and 90 cm. Secondly, temperature's response intensity is 0.2004, 0.0163, and 0.0035 °C for 1% variation in moisture, and moisture's response intensity is 0.0638%, 0.0163%, and 0.0050% for 1 °C variation in temperature at 10, 30, and 90 cm. Thirdly, the past value of soil moisture helps predict soil temperature at 10, 30, and 90 cm. Besides, the past value of soil temperature helps predict soil moisture at 10 and 30 cm, but not at 90 cm. We verified this model by using data from a different year and linking it to soil plant atmospheric continuum model.

## INTRODUCTION

The hydrothermal conditions of a wheat-soil system are essential to the dynamic balance of heat, moisture, and organic matter within the entire system and the thriving of wheat (*Yang, Shang & Guan, 2012*; *Sun et al., 2018*). Because of complicated biological, physical, and chemical processes like soil respiration, soil evaporation, plant transpiration, and so on, soil temperature and moisture have a close dynamic relationship with each other. Therefore, in order to better control and predict the hydrothermal conditions of a

Corresponding author
Xiaohang Feng,
1600018809@pku.edu.cn

wheat-soil system, models simulating soil moisture-heat coupling need to be established (*Perez, Paez & Figueroa, 2013*; *Whelan et al., 2015*).

Most of the models employed to simulate moisture-heat coupling of soil are mechanistic models which call for complex parameters on soil properties and neglect time stochasticity. The following are some examples. *Philip & De Vries (1957)* took soil evaporation under different temperature into consideration and brought up the theory of moisture-gas-heat coupling transport under mass and energy balance. However, they did not consider the time lag effect and temporal heterogeneity. Based on the Philip model, *Nassar & Horton (1989)* used the water, heat, and salt transport equations which are based on Darcy's, Fourier's, and Fick's laws to establish a model for the coupled transport of water, heat, and solute. But the model required a bunch of complicated parameters. Based on the theoretical integrity, *Liu, Liu & Peng (2005)* established a model for describing the migration of heat, moisture, and gas in arid surface porous soil composed of a wet unsaturated layer and a dry but saturated layer. *Bittelli et al. (2008)* established a fully coupled numerical model to solve the governing equations for liquid water, water vapor, and heat transfer in bare soil. *Whelan et al. (2015)* studied the impact of temperature and moisture on soil water repellency by designing and conducting experiments and factorial ANOVAs. However, the study fails to further explore characteristics like time lag and intensity of temperature and moisture's influence on each other. *Lu & Dong (2015)* set up a closed-formed equation for the thermal conductivity of unsaturated soils and shows that the soil water retention curve can be used to predict the thermal conductivity of sands. *Striegl & Loheide (2012)* developed a distributed soil moisture sensing system that addressed the difficulty of characterizing both spatial and temporal soil moisture dynamics at site scales. However, under wet conditions insensitivity of the instrument response curve adversely affected accuracy. *Steele-Dunne et al. (2010)* used distributed temperature sensing to obtain simultaneous measurements of soil moisture over large areas, but they fail to address the complexity of deriving soil moisture due to the uncertainty and non-uniqueness in the relationship between thermal conductivity and soil moisture.

Most of the models mentioned above are deterministic. Since in reality soil moisture and soil temperature constantly vary with disturbances from an assortment of factors like weather and soil, stochastic models are more accurate in terms of prediction than deterministic models (*Bolin, Wallin & Lindgren, in press*). Moreover, mechanistic models mentioned above require a lot of parameters which are difficult to obtain in some cases and some models can only be applied to a certain situation which is quite limited (*Pan et al., 2018*). In comparison, time series models prove to have a wider application since they can be easily established even if nothing but the hydrological time series data are in hand. In addition, although the hysteresis effect has been included in previous models like soil plant atmospheric continuum (SPAC), few indices have been established to quantify the lagging effect.

To take random variation into consideration, and to deal with the situation when only time series data are in hand, we designed a stochastic modeling technique for accurately analyzing moisture-heat coupling within a wheat-soil system. The model consists of

three two-dimensional vector autoregression models, impulse response functions, and Granger causality tests.

## METHODS

### Models

#### Vector autoregression model

Vector autoregression model treats all variables as endogenous variables, accounting for Sims' critique that the exogeneity[1] assumptions of some of the variables in simultaneous equations models are ad hoc and often not backed by fully developed theories. Impulse response analysis and Granger causality tests are tools which have been proposed for disentangling the relations between the variables in a vector autoregression model (*Droumaguet, Warne & Wozniak, 2017*; *Bouri et al., 2018*). The process of establishing a vector autoregression system is as follows.

First, we remove the seasonality of the time series $x_t$, $t = 1, 2, \ldots, n$. Seasonality in a time series is a regular pattern of changes that repeats over $S$ time periods, where $S$ defines the number of time periods until the pattern repeats again. In this case, $S = 24$ (hours per day) is the span of the periodic seasonal behavior. Seasonal differencing is defined as a difference between a value and a value with lag that is a multiple of $S$. With $S = 24$, a seasonal difference is $y_t = (1 - B^{24})x_t = x_t - x_{t-24}$, $t = 25, 26, \ldots, n$, where $B$ is the lag operator.

Next, we check the stationarity of the time series after seasonal adjustment, because average values of non-stationary time series cannot be used if the time period is not set since they are influenced by changes in time. A stochastic process is called stationary when the average and variance values are constant in the corresponding period and covariance values between any two time points depend not on the specific time point but on the lag between these two time points. The three conditions stated above for a time series $\{y_t\}$ to be stationary can be expressed in the following way:

$$E(y_t) = \mu, \text{Var}(y_t) = \sigma^2, \text{Cov}(y_t, y_{t-k}) = \gamma_k \tag{1}$$

where $\mu$, $\sigma^2$, and $\gamma$ are average, variance, and auto-covariance, respectively.

If the time series has a unit root, then it is not stationary. Under such circumstances, we restore the stationarity of the time series by making a difference of it. We use the augmented Dickey–Fuller (ADF) test method to check the stationarity of the time series by testing whether the time series has a unit root. There are three model variants corresponding to three couples of hypothesis in the ADF test.

The first is an autoregressive model variant (referred to as AR), which specifies a test of the null model:

$$H_0: y_t = y_{t-1} + \beta_1 \Delta y_{t-1} + \beta_2 \Delta y_{t-2} + \cdots + \beta_p \Delta y_{t-p} + \varepsilon_t \tag{2}$$

against the alternative model:

$$H_1: y_t = \varnothing y_{t-1} + \beta_1 \Delta y_{t-1} + \beta_2 \Delta y_{t-2} + \cdots + \beta_p \Delta y_{t-p} + \varepsilon_t \tag{3}$$

with $\varnothing < 1$, where $\beta_1, \beta_2, \ldots, \beta_p$ are regression coefficients, and $\varepsilon_t$ is the error term.

[1] Exogenous variables are also called "input variables," thus "exogeneity" describes whether the variable is exogenous or not. An exogenous variable is completely determined by the external part of the system and are input into the system. It only affects the system and is not affected by the system.

The second is an autoregressive model with drift variant (referred to as ARD), which specifies a test of the null model:

$$H_0: y_t = y_{t-1} + \beta_1 \Delta y_{t-1} + \beta_2 \Delta y_{t-2} + \cdots + \beta_p \Delta y_{t-p} + \varepsilon_t \tag{4}$$

against the alternative model:

$$y_t = c + \varnothing y_{t-1} + \beta_1 \Delta y_{t-1} + \beta_2 \Delta y_{t-2} + \cdots + \beta_p \Delta y_{t-p} + \varepsilon_t \tag{5}$$

with $\varnothing < 1$, where $c$ is the constant term, $\beta_1, \beta_2, \ldots, \beta_p$ are regression coefficients, and $\varepsilon_t$ is the error term.

The third is a trend-stationary model variant (referred to as TS), which specifies a test of the null model:

$$H_0: y_t = c + y_{t-1} + \beta_1 \Delta y_{t-1} + \beta_2 \Delta y_{t-2} + \cdots + \beta_p \Delta y_{t-p} + \varepsilon_t \tag{6}$$

against the alternative model:

$$H_1: y_t = c + \delta t + \varnothing y_{t-1} + \beta_1 \Delta y_{t-1} + \beta_2 \Delta y_{t-2} + \cdots + \beta_p \Delta y_{t-p} + \varepsilon_t \tag{7}$$

with $\varnothing < 1$, where $c$ is the constant term, $\delta$ is the coefficient of the trend term, $\beta_1, \beta_2, \ldots, \beta_p$ are regression coefficients, and $\varepsilon_t$ is the error term (*Chen et al., 2018*).

Then, we have to determine the optimal lag order for the vector autoregression model. The proper selection of lag is important because long lag structures reduce the error term's correlation yet they may lack efficiency. We use the estimators Akaike information criterion (AIC) and Bayesian information criterion (BIC), both of which are founded on information theory and estimate relative goodness of fit of given models, as standards of choosing the optimal lag of the vector autoregression model (*Mao & Shang, 2018*).

AIC value of a model is calculated as follows:

$$\text{AIC} = 2k - 2\ln\left(\hat{L}\right) \tag{8}$$

where $k$ is the number of estimated parameters in the model and $\hat{L}$ is the maximum value of the likelihood function for the model.

BIC value of a model is calculated as follows:

$$\text{BIC} = \ln(n)k - 2\ln\left(\hat{L}\right) \tag{9}$$

where $n$ is the number of data points in $x$ or the number of observations.

When fitting the models, it is possible to increase the simulation accuracy by adding parameters, but doing so may result in overfitting. Both BIC and AIC attempt to solve this problem by introducing a penalty term for the number of parameters in the model. And the lag order corresponding to the lowest AIC value or BIC value is taken as the optimal lag order. We use maximum likelihood estimation[2] to estimate the model parameters.

The stability test of the established model is required because if the model is unstable, some results will not be valid (such as the standard error of the impulse response function). In this paper, we use the AR root test and if all the roots of the vector autoregression model estimated based on empirical data have a reciprocal of less than 1 (i.e., if they are within the unit circle), they are stable.

---

[2] The method of maximum likelihood is based on the likelihood function. Suppose we are given a family of distributions $\{f(\cdot; \theta) | \theta \in \Theta\}$, where $\theta$ denotes the parameters (possibly multi-dimensional) for the model. The method defines a maximum likelihood estimate:

$$\hat{\theta} \in \left\{ \underset{\theta \in \Theta}{\arg\max} \, \mathcal{L}(\theta; x) \right\},$$ where $\mathcal{L}(\theta; x)$

denotes the likelihood function, because intuitively this selects parameter values that make the data most probable.

The way to check whether a vector autoregression model is stationary is as follow. For vector autoregression model

$$(y_t - \mu) = \phi_1(y_{t-1} - \mu) + \phi_2(y_{t-2} - \mu) + \cdots + \phi_p(y_{t-p} - \mu) + \varepsilon_t$$

where $y_t$ is a vector in $R^n$, we first write it in the form of deviation. Then we define matrix $F$:

$$F = \begin{bmatrix} \phi_1 & \phi_2 & \phi_3 & \cdots & \phi_{p-1} & \phi_p \\ I_n & 0 & 0 & \cdots & 0 & 0 \\ 0 & I_n & 0 & \cdots & 0 & 0 \\ \vdots & \vdots & \vdots & \cdots & \vdots & \vdots \\ 0 & 0 & 0 & \cdots & I_n & 0 \end{bmatrix}$$

where $I_n$ is a unit matrix of $n$ dimension. If all characteristic roots of matrix $F$ fall within a unit circle, then the vector autoregression model is stable.

### Impulse response function

The impulse response function is used to characterize the influence of a standard deviation shock of random perturbation terms on the current and future values of other variables. It can visually describe the interactions and effects between variables (*Schoukens, Godfrey & Schoukens, 2018*). We take vector autoregression that has one lag order and contains tow variables ($y$ and $z$) as an example. The matrix form of structure a vector autoregression model can be written as follow.

$$\begin{bmatrix} 1 & b_{12} \\ b_{21} & 1 \end{bmatrix} \begin{bmatrix} y_t \\ z_t \end{bmatrix} = \begin{bmatrix} b_{10} \\ b_{20} \end{bmatrix} + \begin{bmatrix} \gamma_{11} & \gamma_{12} \\ \gamma_{21} & \gamma_{22} \end{bmatrix} \begin{bmatrix} y_{t-1} \\ z_{t-1} \end{bmatrix} + \begin{bmatrix} \varepsilon_{yt} \\ \varepsilon_{zt} \end{bmatrix} \tag{10}$$

Then we write it in the standard form as follow.

$$\begin{bmatrix} y_t \\ z_t \end{bmatrix} = \begin{bmatrix} a_{10} \\ a_{20} \end{bmatrix} + \begin{bmatrix} a_{11} & a_{12} \\ a_{21} & a_{22} \end{bmatrix} \begin{bmatrix} y_{t-1} \\ z_{t-1} \end{bmatrix} + \begin{bmatrix} e_{1t} \\ e_{2t} \end{bmatrix} \tag{11}$$

Moreover, we can write it in the following form if the vector autoregression model is stable.

$$\begin{bmatrix} y_t \\ z_t \end{bmatrix} = \begin{bmatrix} \bar{y}_t \\ \bar{z}_t \end{bmatrix} + \sum_{i=0}^{\infty} \begin{bmatrix} a_{11} & a_{12} \\ a_{21} & a_{22} \end{bmatrix}^i \begin{bmatrix} e_{1t-i} \\ e_{2t-i} \end{bmatrix} \tag{12}$$

For the transformation from (10) to (11), the error term can be transformed as follow.

$$\begin{bmatrix} e_{1t} \\ e_{2t} \end{bmatrix} = 1/(1 - b_{12}b_{21}) \begin{bmatrix} 1 & -b_{12} \\ -b_{21} & 1 \end{bmatrix} \begin{bmatrix} \varepsilon_{yt} \\ \varepsilon_{zt} \end{bmatrix} \tag{13}$$

By combining Eqs. (12) and (13), we obtain

$$\begin{bmatrix} y_t \\ z_t \end{bmatrix} = \begin{bmatrix} \bar{y}_t \\ \bar{z}_t \end{bmatrix} + 1/(1 - b_{12}b_{21}) \sum_{i=0}^{\infty} \begin{bmatrix} a_{11} & a_{12} \\ a_{21} & a_{22} \end{bmatrix}^i \begin{bmatrix} 1 & -b_{12} \\ -b_{21} & 1 \end{bmatrix} \begin{bmatrix} \varepsilon_{yt-i} \\ \varepsilon_{zt-i} \end{bmatrix} \tag{14}$$

In order to simplify the equation, we define matrix $\phi_i$, and denote its elements as $\phi_{jk}(i)$.

$$\phi_i = A_1^i/(1 - b_{12}b_{21}) \begin{bmatrix} 1 & -b_{12} \\ -b_{21} & 1 \end{bmatrix} \tag{15}$$

Thus, the moving average of the Eqs. (11) and (12) can be expressed as

$$\begin{bmatrix} y_t \\ z_t \end{bmatrix} = \begin{bmatrix} \bar{y}_t \\ \bar{z}_t \end{bmatrix} + \sum_{i=0}^{\infty} \begin{bmatrix} \phi_{11}(i) & \phi_{12}(i) \\ \phi_{21}(i) & \phi_{22}(i) \end{bmatrix} \begin{bmatrix} \varepsilon_{yt-i} \\ \varepsilon_{zt-i} \end{bmatrix} \tag{16}$$

Moving average is a useful tool for studying the mutual influence between the two series $\{y_t\}$ and $\{z_t\}$, for the impulse response function of $\{y_t\}$ and $\{z_t\}$ to one unit shock of $\varepsilon_{yt}$ and $\varepsilon_{zt}$ can be built using the coefficients of $\phi_i$. Here, $\phi_{11}(i)$, $\phi_{12}(i)$, $\phi_{21}(i)$, and $\phi_{22}(i)$ are called impulse response functions. And the graphs of these functions show the impulse response of the time series to one unit of positive shock.

To be more specific, coefficient $\phi_{12}(0)$ indicates the current impact of one unit positive change in $\varepsilon_{zt}$ have on $y_t$. Similarly, after one period of calibration, $\phi_{11}(1)$ and $\phi_{12}(1)$ also denote the impact of one unit positive change in $\varepsilon_{yt}$ and $\varepsilon_{zt}$ have on $y_{t+1}$. The cumulative effect of $\varepsilon_{yt}$ and $\varepsilon_{zt}$'s unit impulse response can be obtained by the appropriate add up of the coefficients of the impulse response functions. For instance, after $n$ periods the impact of $\varepsilon_{zt}$ on $y_{t+n}$ is $\phi_{12}(n)$. Therefore, after $n$ periods the accumulative impact of $\varepsilon_{zt}$ on $y_t$ is $\sum_{i=0}^{n} \phi_{12}(i)$. If $n$ tends to be positive infinite, the accumulative effect can be obtained. Since we have assumed that $y_t$ and $z_t$ are stationary, when $j$ and $k$ approximate infinite, the value of $\phi_{jk}(i)$ is 0. Since all parameters of the vector autoregression system can be calculated, it is totally possible to track every value of $\varepsilon_{yt}$ and $\varepsilon_{zt}$'s impact on $\{y_t\}$ and $\{z_t\}$ can be calculated. The time when $\phi_{jk}(i)$ approximates 0 is the time lag of impulse response, and the gap between the positive largest value and negative largest value of $\phi_{jk}(i)$ can be used to describe the intensity of the impulse response.

### Granger causality test

The Granger causality test is used to test if the variable $y$ can be used to predict the variable $x$. That is, when the variable $x$ is regressed according to the past value of the variable $y$, the explanatory ability of the regression can be significantly enhanced. It should be noted that two variables have a temporal "causal relationship" but do not necessarily have a logical causal relationship.

In an attempt to test whether past values of $y$ help predict $x$, we first perform an ordinary least squares estimation:

$$x_t = c_1 + a_1 x_{t-1} + a_2 x_{t-2} + \cdots + a_p x_{t-p} + b_1 y_{t-1} + b_2 y_{t-2} + \cdots + b_p x_{t-p} + u_t \tag{17}$$

And we propose the null hypothesis:

$$H_0: b_1 = b_2 = \cdots = b_p = 0 \tag{18}$$

which means that $y$ is not the Granger reason for $x$. To conduct an $F$-test for this hypothesis, first, we estimate the equation without restriction $H_0$, and obtain the sum of residual:

$$RSS_0 = \sum_{t=1}^{T} \hat{u}_t^2 \tag{19}$$

Then, we estimate the equation with restriction $H_0$, and obtain the sum of residual:

$$\text{RSS}_1 = \sum_{t=1}^{T} \hat{e}_t^2 \tag{20}$$

We build the $F$ statistic using the expression below:

$$\frac{(\text{RSS}_1 - \text{RSS}_0)/p}{\text{RSS}_0/(T - 2p - 1)} \sim F(p, T - 2p - 1) \tag{21}$$

where $T$ is the size of the sample. Based on the data, if the value of $F$ statistic is higher than the critical value, then we reject the null hypothesis and admit that increasing the lag order of $y$ can significantly improve the explanatory ability of the model. Under such circumstances, we say that the past value of y helps predict $x$ (*Stokes & Purdon, 2018*).

### Stochastic model system for soil moisture-heat coupling

First, we build three vector autoregression models at the depth of 10, 30, and 90 cm. We choose these three depths because, during the greening period which is around 15th March, the diameter of wheat root reaches its maximum approximately at the depth of 10, 30, and 90 cm, respectively, thus resulting in more water absorption at these three depths. In addition, usually, the soil is divided into topsoil, subsoil, and substratum, the representative depths of which approximate 10, 30, and 90 cm. In addition, we select hourly data on soil moisture and temperature from December to May because this period covers the main growing stages of winter wheat. This time period can be adjusted according to the growing stages of different wheat species (*Awad et al., 2018*).

Second, we build six impulse response functions for these three vector autoregression systems. Based on the results of impulse response function analysis, we are able to obtain the time lag and intensity of impulse response. For example, if one unit of positive shock is given to soil temperature at the depth of 10 cm, we calculate the time length during which obvious deviation from zero can be seen in soil moisture's response to $1\,°C$ of positive variation. This provides us with information on how long soil moisture will be influenced if there is a sudden change in temperature. In addition, the gap between the highest point and lowest point of the response is the intensity of impulse response. The larger the intensity, the stronger the influence of soil temperature has on moisture.

Third, we conduct six Granger causality tests between two variables of soil moisture and temperature at three depths. According to the result of the Granger causality test, we are able to decide a more proper way of forecasting soil moisture and temperature by taking the Granger causal relationship of moisture and temperature into consideration. If, for example, at the depth of 10 cm the past value of soil temperature helps predict soil moisture, it means that the past value of soil temperature influences soil moisture. Therefore, when establishing empirical models forecasting soil moisture, we should choose the ARMAX model which is an ARMA model with an exogenous variable in an attempt to take into consideration of temperature's influence on moisture. On the other hand, if the past value of soil temperature does not help predict soil moisture, it means that the past

value of soil temperature does not have a significant influence on soil moisture. If soil temperature and moisture are Granger reasons for each other, it suggests that the present value of two variables are related to each other's past value. Thus, a vector autoregression model treating all variables as endogenous variables will be the optimal choice.

## Field experiment on soil moisture and temperature

[3] Bainong Dwarf Anti-floating No.58 is a new epoch-making wheat variety developed in 2003 by the Wheat Breeding Center in Henan University of Science and Technology. This variety belongs to the semi-winter mid-maturing variety. The plant height is about 70 cm, with high resistance to lodging and good fullness. It has high resistance to powdery mildew, stripe rust, leaf blight, and medium resistance to sheath blight, strong root activity, good ripening, and yellowing. It also has the preponderance of high yield with the average yield being 7,500–8,250 kg/ha and the maximum being 10,500 kg/ha.

The experiment on the wheat of Bainong Dwarf Anti-floating No. 58[3] was carried out at Shangqiu Experimental Station of Farmland Irrigation Institute of Chinese Academy of Agricultural Sciences with a latitude of 34°35.222′N, longitude of 115°34.515′E, and elevation of 55.6 m. The average annual temperature there is 13.9 °C, and the frost-free period is 180 to 230 days. In addition, the average annual precipitation is 708 mm with the precipitation from July to September accounting for 65–75% of the total annual precipitation, and the average annual evaporation being 1,735 mm. The soil of the test site is a loam, which refers to soil consists of clay (30–40%), silt (30–40%), and sand (30–40%).

The Irrigation method adopted is drip irrigation at the depth of 30 cm under soil surface. The sensor we used for soil water monitoring is called "Soil-Water," which was produced in Australia. The basic components of this soil monitoring system are data collectors, solar modules, mounting components, and soil sensors. The accuracy for temperature is <0.1 °C, and but the accuracy of soil moisture sensor was not provided by the manufacturer. We installed the instrument probes of the moisture meter at the depth of 10, 20, 30, 40, 50, 60, 70, 80, 90, and 100 cm, and measured soil moisture and soil temperature at these depths simultaneously. The instrument's temperature measurement range is from −20 to 60 °C.

Among these 10 sets of data, we select hourly data on soil moisture and soil temperature from December 19th, 2017 to May 30th, 2018 at the depth of 10, 30, and 90 cm, which is in accordance with the model designed above. In addition, the selection of the measurement interval takes into account both the memory capacity of the instrument and the minimum scale of moisture transport, since one-hour interval is sufficient to show the footprint of water movement.

## RESULTS

Figure 1 shows data on soil moisture and temperature measured at the depth of 10, 20, 30, 40, 50, 60, 70, 80, 90, and 100 cm. We can see that both soil temperature and soil moisture content exhibits periodical variations within a day and trends changing with the alternation of seasons. Soil temperature decreases with the cooling trend of climate from December to January and increases with the overall warming of the climate from February to May in next year. Soil moisture content, however, experiences a sudden hike around 2nd March and 19th March from 10 to 50 cm due to irrigation, and gradually decreases from March to May due to evapotranspiration, indicating an active water absorption process of winter wheat root system. Soil moisture content exhibits obvious differences among different depths.

We use hourly data from December 19th, 2017 to May 30th, 2018 on soil moisture content at the soil depths of 10, 30, and 90 cm (hereinafter referred to as *MC10, MC30,* and

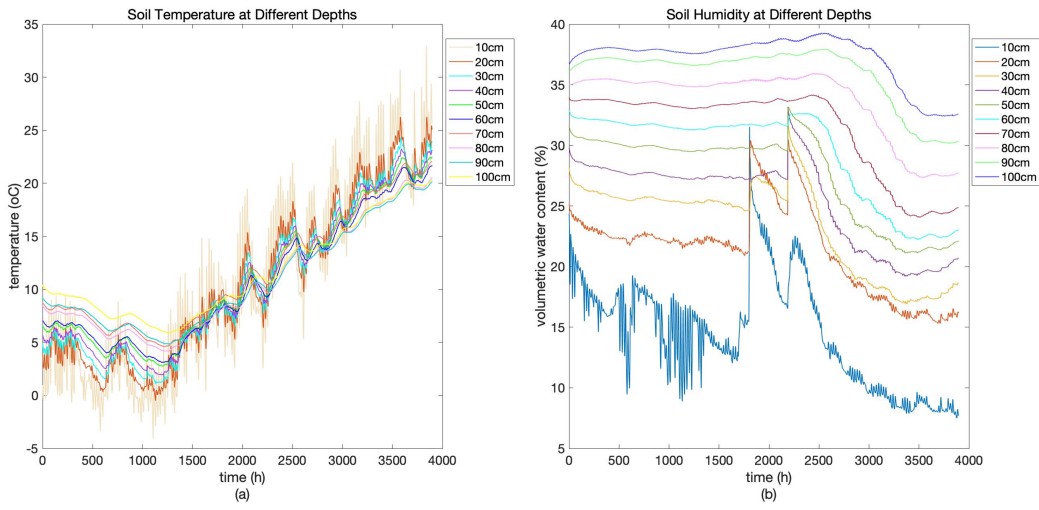

**Figure 1 Soil temperature (A) and soil humidity (B) at different depths.** Soil temperature and soil humidity at the depth of 10–100 cm obtained at the Shangqiu test site, China.

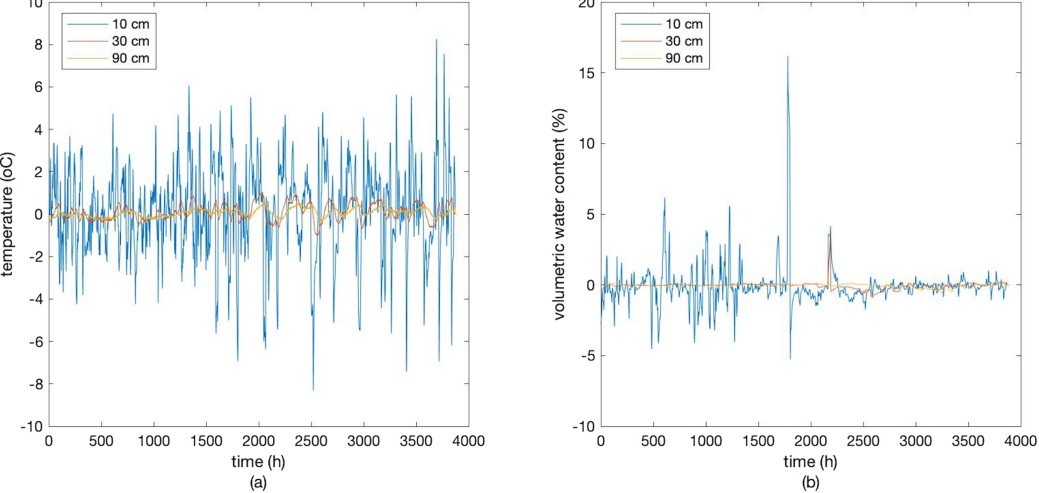

**Figure 2 Deseasonalized soil temperature (A) and moisture (B) at the depth of 10, 30, and 90 cm.** Soil temperature and moisture after deseasonalization at the depth of 10 cm, 30 cm, and 90 cm.

*MC90*, respectively), soil temperature at the depth of 10, 30, and 90 cm (hereinafter referred to as *ST10, ST30*, and *ST90*, respectively).

First, we remove the seasonality of the time series of *MC10, ST10, MC30, ST30, MC90*, and *ST90*, since it is known from the above figures that the six hourly time series have obvious diurnal pattern due to the periodical variations of meteorological factors, evapotranspiration, and root water uptake within a day, especially in shallow layers. After seasonal difference, the period of the data is from December 20th, 2017 to May 30th, 2018. The corresponding time series of the six detrended variables are referred to as *MCSD10, STSD10, MCSD30, STSD30, MCSD90,* and *STSD90*, respectively (Fig. 2).

**Table 1  Unit root test results of the deseasonalized soil moisture and temperature time series.**

| Variable | T stat.[1] | Model[2] | P-value[3] | 1%C.V.[4] | 5%C.V. | 10%C.V. | Conclusion |
|---|---|---|---|---|---|---|---|
| MCSD10 | −11.273 | AR(8) | 0.001*** | −2.568 | −1.942 | −1.617 | Stationary[5] |
| STSD10 | −11.689 | AR(5) | 0.001*** | −2.568 | −1.942 | −1.617 | Stationary |
| MCSD30 | −8.590 | AR(3) | 0.001*** | −2.568 | −1.942 | −1.617 | Stationary |
| STSD30 | −7.423 | AR(8) | 0.001*** | −2.568 | −1.942 | −1.617 | Stationary |
| MCSD90 | −3.927 | AR(8) | 0.001*** | −2.568 | −1.942 | −1.617 | Stationary |
| STSD90 | −2.055 | AR(8) | 0.039** | −2.568 | −1.942 | −1.617 | Stationary |

**Notes:**
[1] T stat. is the t-statistic for performing ADF test.
[2] Model indicates the type of hypothesis selected and the lag period which is added to make the residual become white noise.
[3] P-Value represents the corresponding P-value.
[4] C.V. indicates critical value.
[5] Stationary indicates that the time series is stable.
** Indicates a significant level of 5%.
*** Indicates a significant level of 1%.

According to the above results of the ADF test (Table 1), the variables of *MCSD10, STSD10, MCSD30, STSD30,* and *MCSD90* are all stationary under the significance level of 0.01. The variable of STSD90 is stationary under the significance level of 0.05. Thus, it is appropriate to use the variables of *MCSD10, STSD10, MCSD30, STSD30, MCSD90,* and *STSD90* to build vector autoregression models since they are all stationary.

## Vector autoregression model establishment

The values of AIC and BIC for vector autoregression models of different lags are calculated (Table 2) to determine the optimal lag orders for the vector autoregression models.

For soil depth of 10 cm, among the eight lag orders, lag order 8 has the lowest AIC value and lag order 5 has the lowest BIC value. Since model with a lag order of 5 has fewer parameters to estimate which leads to smaller estimation error for the whole model, we choose 5 as the optimal lag order (Table 2). Based on the optimal lag structure and the time series of *MCSD10* and *STSD10*, we estimate the parameters of the vector autoregression model using maximum likelihood estimation. Most of the t-statistic of the estimated coefficients in the above regression model is significant at the 10% significance level. Although some of the coefficients are not significant, it may because multiple hysteresis values with the same variables in the same equation result in multiple collinearities. Thus, finally, we obtained the estimated vector autoregression model:

$$
\begin{bmatrix} MCSD10_t \\ STSD10_t \end{bmatrix} = \begin{bmatrix} -0.002 \\ 0.005 \end{bmatrix} + \begin{bmatrix} 1.593 & -0.026 \\ 0.167 & 1.754 \end{bmatrix} \begin{bmatrix} MCSD10_{t-1} \\ STSD10_{t-1} \end{bmatrix}
$$
$$
+ \begin{bmatrix} -0.631 & 0.019 \\ -0.205 & -0.816 \end{bmatrix} \begin{bmatrix} MCSD10_{t-2} \\ STSD10_{t-2} \end{bmatrix} + \begin{bmatrix} 0.274 & 0.012 \\ -0.061 & 0.026 \end{bmatrix} \begin{bmatrix} MCSD10_{t-3} \\ STSD10_{t-3} \end{bmatrix} \quad (22)
$$
$$
+ \begin{bmatrix} -0.526 & 0.024 \\ -0.094 & -0.060 \end{bmatrix} \begin{bmatrix} MCSD10_{t-4} \\ STSD10_{t-4} \end{bmatrix} + \begin{bmatrix} 0.257 & -0.031 \\ 0.008 & 0.068 \end{bmatrix} \begin{bmatrix} MCSD10_{t-5} \\ STSD10_{t-5} \end{bmatrix} + \begin{bmatrix} u_{1t} \\ u_{2t} \end{bmatrix}
$$

For soil depth of 30 cm, among the eight lag orders, lag order 8 has the lowest AIC value and the lowest BIC value. Thus, we choose 8 as the optimal lag order and build a vector

**Table 2 Calculated In(L), AIC, and BIC for vector auto-regression models of different lags at the depth of 10, 30, and 90 cm.**

| Lag | 10 cm | | | 30 cm | | | 90 cm | | |
|---|---|---|---|---|---|---|---|---|---|
| | $\ln(\hat{L})$ | AIC | BIC | $\ln(\hat{L})$ | AIC | BIC | $\ln(\hat{L})$ | AIC | BIC |
| 1 | −2,298.5 | 4,609.1 | 4,646.6 | 11,500.7 | −22,989.4 | −22,951.8 | 19,089.2 | −38,166.3 | −38,128.8 |
| 2 | 242.7 | −465.5 | −402.9 | 11,758.8 | −23,497.7 | −23,435.1 | 19,625.2 | −39,230.3 | −39,167.7 |
| 3 | 277.9 | −527.8 | −440.1 | 11,762.2 | −23,496.5 | −23,408.8 | 19,814.5 | −39,600.9 | −39,513.3 |
| 4 | 327.8 | −619.6 | −507.0 | 11,826.4 | −23,616.9 | −23,504.2 | 19,927.4 | −39,818.8 | −39,706.1 |
| 5 | 478.2 | −912.4 | −774.7 | 11,942.4 | −23,840.7 | −23,703.0 | 20,013.4 | −39,982.9 | −39,845.1 |
| 6 | 493.0 | −934.1 | −771.3 | 12,050.5 | −24,048.9 | −23,886.2 | 20,035.8 | −40,019.7 | −39,856.9 |
| 7 | 493.2 | −926.4 | −738.6 | 12,102.0 | −24,144.1 | −23,956.3 | 20,054.4 | −40,048.7 | −39,860.9 |
| 8 | 508.4 | −948.8 | −736.0 | 12,143.2 | −24,218.3 | −24,005.5 | 20,068.5 | −40,069.0 | −39,856.2 |

Note:
The first row of the table denotes the possible lag orders of the model. The table is used to choose the optimal lag order for each model by comparing the value In(L), AIC, and BIC.

autoregression model (Table 2). We obtained the estimated vector autoregression model as follow:

$$
\begin{bmatrix} MCSD30_t \\ STSD30_t \end{bmatrix} = \begin{bmatrix} -0.002 \\ 0.001 \end{bmatrix} + \begin{bmatrix} 1.182 & 0.117 \\ -0.006 & 0.559 \end{bmatrix} \begin{bmatrix} MCSD30_{t-1} \\ STSD30_{t-1} \end{bmatrix}
$$
$$
+ \begin{bmatrix} -0.173 & -0.090 \\ -0.022 & 0.344 \end{bmatrix} \begin{bmatrix} MCSD30_{t-2} \\ STSD30_{t-2} \end{bmatrix} + \begin{bmatrix} -0.016 & 0.020 \\ 0.042 & 0.228 \end{bmatrix} \begin{bmatrix} MCSD30_{t-3} \\ STSD30_{t-3} \end{bmatrix}
$$
$$
+ \begin{bmatrix} -0.002 & -0.054 \\ -0.003 & 0.149 \end{bmatrix} \begin{bmatrix} MCSD30_{t-4} \\ STSD30_{t-4} \end{bmatrix} + \begin{bmatrix} 0.000 & 0.013 \\ -0.022 & 0.012 \end{bmatrix} \begin{bmatrix} MCSD30_{t-5} \\ STSD30_{t-5} \end{bmatrix} \quad (23)
$$
$$
+ \begin{bmatrix} -0.006 & -0.014 \\ 0.019 & -0.074 \end{bmatrix} \begin{bmatrix} MCSD30_{t-6} \\ STSD30_{t-6} \end{bmatrix} + \begin{bmatrix} 0.012 & 0.040 \\ -0.010 & -0.082 \end{bmatrix} \begin{bmatrix} MCSD30_{t-7} \\ STSD30_{t-7} \end{bmatrix}
$$
$$
+ \begin{bmatrix} -0.027 & -0.029 \\ -0.002 & -0.148 \end{bmatrix} \begin{bmatrix} MCSD30_{t-8} \\ STSD30_{t-8} \end{bmatrix} + \begin{bmatrix} u_{1t} \\ u_{2t} \end{bmatrix}
$$

For soil depth of 90 cm, among the eight lag orders, lag order 8 has the lowest AIC value and lag order 7 has the lowest BIC value. Since model with a lag order of 7 has fewer parameters to estimate which leads to smaller estimation error for the whole model, we choose 7 as the optimal lag order (Table 2). We obtained the estimated vector autoregression model as follow:

$$
\begin{bmatrix} MCSD90_t \\ STSD90_t \end{bmatrix} = \begin{bmatrix} 0.690 & 0.058 \\ 0.133 & 0.345 \end{bmatrix} \begin{bmatrix} MCSD90_{t-1} \\ STSD90_{t-1} \end{bmatrix} + \begin{bmatrix} 0.336 & 0.025 \\ 0.026 & 0.200 \end{bmatrix} \begin{bmatrix} MCSD90_{t-2} \\ STSD90_{t-2} \end{bmatrix}
$$
$$
+ \begin{bmatrix} 0.130 & -0.001 \\ 0.045 & 0.191 \end{bmatrix} \begin{bmatrix} MCSD90_{t-3} \\ STSD90_{t-3} \end{bmatrix} + \begin{bmatrix} -0.024 & -0.010 \\ 0.079 & 0.103 \end{bmatrix} \begin{bmatrix} MCSD90_{t-4} \\ STSD90_{t-4} \end{bmatrix}
$$
$$
+ \begin{bmatrix} -0.061 & -0.031 \\ -0.166 & 0.039 \end{bmatrix} \begin{bmatrix} MCSD90_{t-5} \\ STSD90_{t-5} \end{bmatrix} + \begin{bmatrix} -0.004 & -0.008 \\ -0.132 & 0.056 \end{bmatrix} \begin{bmatrix} MCSD90_{t-6} \\ STSD90_{t-6} \end{bmatrix} \quad (24)
$$
$$
+ \begin{bmatrix} -0.071 & -0.032 \\ 0.011 & 0.058 \end{bmatrix} \begin{bmatrix} MCSD90_{t-7} \\ STSD90_{t-7} \end{bmatrix} + \begin{bmatrix} u_{1t} \\ u_{2t} \end{bmatrix}
$$

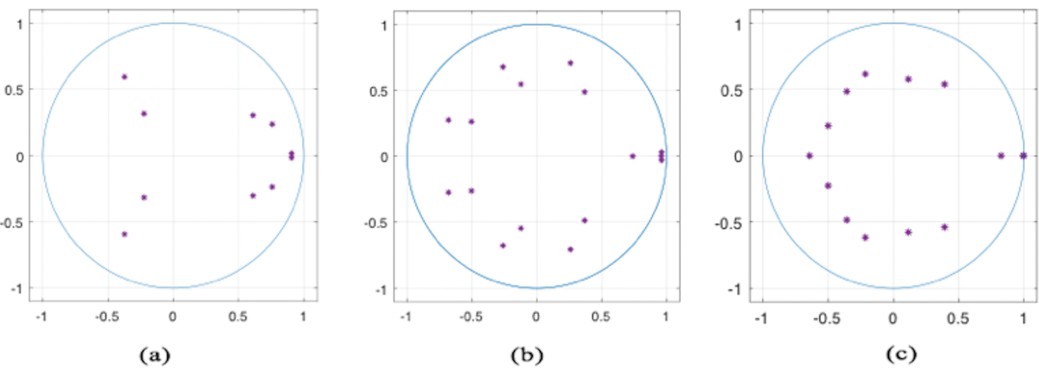

**Figure 3 Inverse roots of AR characteristic polynomial at 10 cm (A), 30 cm (B), and 90 cm (C).** In this complex plane, the horizontal axis is the real axis, and the vertical axis is the imaginary axis. Inverse roots of AR characteristic polynomial at 10 cm (A), 30 cm (B), and 90 cm (C) show that all three models are stationary.

**Table 3 Impulse response results of *MCSD10*, *STSD10*, *MCSD30*, *STSD30*, *MCSD90*, and *STSD90*.**

| RP[1] (h) | Response of *MCSD10* to one unit of positive shock in *STSD10* (%) | Response of *STSD10* to one unit of positive shock in *MCSD10* (°C) | Response of *MCSD30* to one unit of positive shock in STSD30 (%) | Response of *STSD30* to one unit of positive shock in *MCSD30* (°C) | Response of *MCSD90* to one unit of positive shock in *STSD90* (%) | Response of *STSD90* to one unit of positive shock in *MCSD90* (°C) |
|---|---|---|---|---|---|---|
| 0 | 0.0312 | 0.0453 | −0.0077 | −0.0045 | −0.0033 | −0.0107 |
| 1 | 0.0426 | 0.1112 | −0.0046 | −0.0029 | −0.0003 | −0.0024 |
| 2 | 0.0410 | 0.1695 | −0.0051 | −0.0051 | 0.0002 | −0.0018 |
| 3 | 0.0434 | 0.1942 | −0.0033 | −0.0044 | 0.0004 | −0.0015 |
| 4 | 0.0449 | 0.2004 | −0.0033 | −0.0048 | 0.0010 | −0.0000 |
| 5 | 0.0470 | 0.1876 | −0.0019 | −0.0064 | 0.0009 | −0.0008 |
| 6 | 0.0530 | 0.1581 | −0.0011 | −0.0063 | 0.0013 | −0.0014 |
| 7 | 0.0584 | 0.1268 | 0.0011 | −0.0070 | 0.0009 | −0.0009 |
| 8 | 0.0620 | 0.0973 | 0.0019 | −0.0073 | 0.0013 | −0.0006 |
| 9 | 0.0638 | 0.0718 | 0.0025 | −0.0081 | 0.0014 | −0.0006 |
| 10 | 0.0626 | 0.0543 | 0.0036 | −0.0085 | 0.0015 | −0.0005 |
| 11 | 0.0586 | 0.0433 | 0.0045 | −0.0091 | 0.0015 | −0.0004 |
| 12 | 0.0531 | 0.0374 | 0.0054 | −0.0096 | 0.0016 | −0.0005 |

**Note:**
The first column of the table denotes retrospective periods, with hour as the unit. The next six columns display the results of response of *MCSD* to one unit of positive shock in *STSD* (%) and response of *MCSD* to one unit of positive shock in *STSD* (%).
[1] RP represents retrospective periods, with hour as the unit.

Then we check the stability of these three vector autoregression models. Figure 3 shows that all unit roots fall within the unit root circle, so it is reasonable to believe that the vector autoregression models at these three soil depths are stable, indicating that there is a long-term stable relationship between the variables selected which can be further analyzed.

## Impulse response function analysis

The impulse response results at three different depths are shown in Table 3.

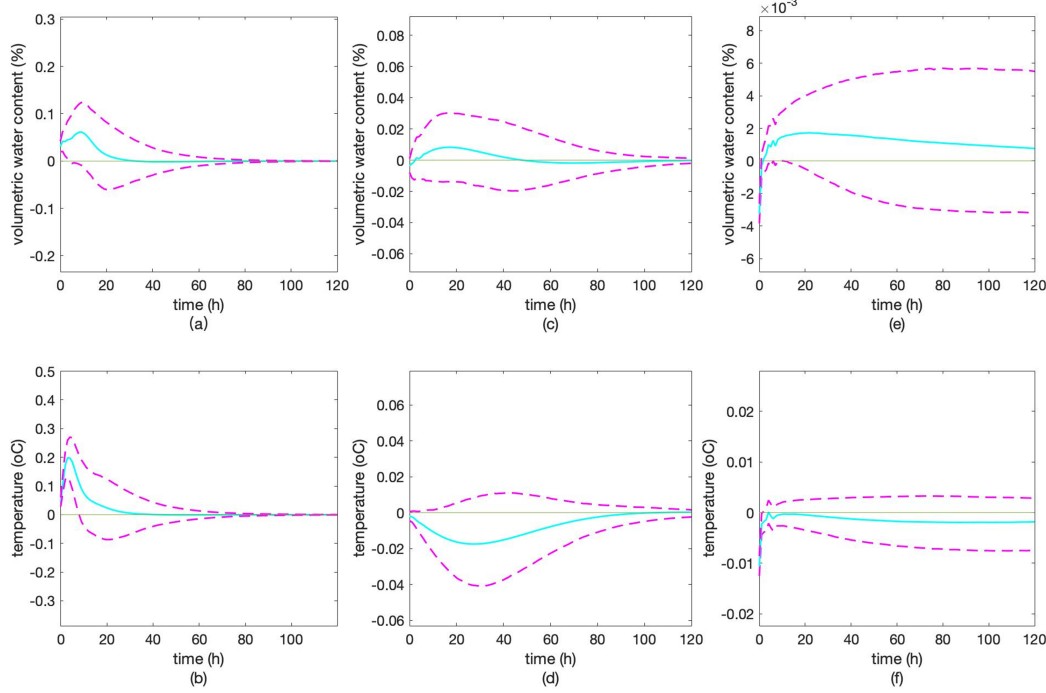

**Figure 4 Impulse response results.** Impulse response of *MCSD10* to shock in *STSD10* (A), impulse response of *STSD10* to shock in *MCSD10* (B), impulse response of *MCSD30* to shock in *STSD30* (C), impulse response of *STSD30* to shock in *MCSD30* (D), impulse response of *MCSD90* to shock in *STSD90* (E), impulse response of *STSD90* to shock in *MCSD90* (F). The solid line of blue fluorescence represents the impulse response and the red dashed line represents the confidence interval of the impulse response with a confidence level of 0.90.           

The impulse response diagrams at three different depths are shown in Fig. 4. The horizontal axis represents the number of retrospective periods from 0 to 120 h, while the vertical axis represents the response of the dependent variable to the shock. The confidence interval is used because the vector autoregression model coefficients have some error in the estimation. Setting the confidence interval can accommodate the inherent uncertainty of the parameters.

When *STSD10* is given one unit of positive shock, *MCSD10* reacts to 0.0312% in the current period, and then monotonically increases until it reaches a positive maximum of 0.0638% in the ninth period. Then, it gradually decreases and falls near zero in the long run, indicating that the vector autoregression system is stable. It can be seen that the impact of *STSD10* on *MCSD10* has a lagging effect since its impulse response is only close to zero after around 25 periods. Moreover, the intensity of the influence which can be quantified as 0.0638% is relatively high (Fig. 4A).

When *MCSD10* is given one unit of positive shock, *STSD10* reacts to 0.0453 °C in the current period, and then monotonically increases until it reaches a positive maximum of 0.2004 °C in the fourth period. Then it gradually decreases and falls near zero in the long run, indicating that the vector autoregression system is stable. It can be seen that the impact of *MCSD10* on *STSD10* has a lagging effect since its impulse response never falls

below zero and is only close to zero after 50 periods. Moreover, the intensity of the influence which can be quantified as 0.2004 °C is quite high (Fig. 4B).

When *STSD30* is given one unit of positive shock, *MCSD30* reacts to −0.0077% in the current period, and then monotonically increases until it reaches a positive maximum of 0.0086% in the 22nd, 23rd, 24th period. Then it gradually decreases and falls near zero in the long run, indicating that the vector autoregression system is stable. Besides, the impact of *STSD30* on *MCSD30* has a lagging effect based on the fact that it is only close to zero after around 50 periods. Moreover, the intensity of the influence which can be quantified as 0.0163% is lower than the intensity of *MCSD10*'s response to *STSD10* (Fig. 4C).

When *MCSD30* is given one unit of positive shock, *STSD30* reacts to −0.0045 °C in the current period, and then monotonically decreases until it reaches a negative maximum of −0.0163 °C at the 36th, 37th, 38th, 39th period. Then it gradually increases and fall near zero in the long run, indicating that the vector autoregression system is stable. It can be seen that the impact of *MCSD30* on *STSD30* has a lagging effect since its impulse response is only close to zero after the 120th period. Moreover, the intensity of the influence which can be quantified as 0.0163 °C is lower than the intensity of *STSD10*'s response to *MCSD10* (Fig. 4D).

When *STSD90* is given one unit of positive shock, *MCSD90* reacts to −0.0033% in the current period, and then monotonically increases until it reaches a positive maximum of 0.0017% from the 17th to 30th period. Then it gradually decreases and falls near zero in the long run, indicating that the vector autoregression system is stable. It can be seen that the impact of *STSD90* on *MCSD90* has a lagging effect based on the fact that it is only close to zero after 120 periods. Moreover, the intensity of the influence which can be quantified as 0.0050% is lower than the intensity of *MCSD30*'s response to *STSD30* (Fig. 4E).

When *MCSD90* is given one unit of positive shock, *STSD90* reacts to −0.0107 °C in the current period, and then monotonically increases and reaches a relatively stable value of −0.0035 °C at around the 120th period. It can be seen that the impact of *MCSD90* on *STSD90* has a lagging effect. Moreover, the intensity of the influence which can be quantified as 0.0035 °C is lower than the intensity of *STSD30*'s response to *MCSD30* (Fig. 4F).

## Granger causality test

By carrying out Granger causality test, we figured out whether the past value of *STSD10* (*STSD30*, *STSD90*) helps predict *MCSD10* (*MCSD30*, *MCSD90*) and whether the past value of *MCSD10* (*MCSD30*, *MCSD90*) helps predict *STSD10* (*STSD30*, *STSD90*).

From the results in Tables 4 and 5, it is known that the past value of *MCSD10* helps predict *STSD10*, but the past value of *STSD10* does not help predict *MCSD10*. This means that the past value of soil moisture content is helpful in terms of predicting the present value of soil temperature at the depth of 10 cm, while soil temperature has a small influence in terms of predicting soil moisture content at the depth of 10 cm. Besides, the results also indicate soil moisture content varies ahead of soil temperature.

**Table 4 Granger causality test determining whether *STSD10* Granger causes *MCSD10*.**

| Significance | F-statistic | Critical value | Conclusion |
|---|---|---|---|
| 0.01 | 0.5694 | 6.6415 | No |
| 0.05 | 0.5694 | 3.8439 | No |
| 0.10 | 0.5694 | 2.7068 | No |

Note:
The table displays the results of the Granger causality test of the depth of 10 cm. By comparing the value of *F*-Statistics and critical values at the significance levels of 0.01, 0.05, and 0.10, we can deduce the Granger causal relationship between soil moisture and temperature.

**Table 5 Granger causality test determining whether *MCSD10* Granger causes *STSD10*.**

| Significance | F-statistic | Critical value | Conclusion |
|---|---|---|---|
| 0.01 | 18.3573 | 3.3240 | Yes |
| 0.05 | 18.3573 | 2.3742 | Yes |
| 0.10 | 18.3573 | 1.9463 | Yes |

Note:
The table displays the results of the Granger causality test of the depth of 10 cm. By comparing the value of *F*-Statistics and critical values at the significance levels of 0.01, 0.05, and 0.10, we can deduce the Granger causal relationship between soil moisture and temperature.

**Table 6 Granger causality test determining whether *STSD30* Granger causes *MCSD30*.**

| Significance | F-statistic | Critical value | Conclusion |
|---|---|---|---|
| 0.01 | 12.7695 | 4.6107 | Yes |
| 0.05 | 12.7695 | 2.9981 | Yes |
| 0.10 | 12.7695 | 2.3040 | Yes |

Note:
The table displays the results of the Granger causality test of the depth of 30 cm. By comparing the value of *F*-Statistics and critical values at the significance levels of 0.01, 0.05, and 0.10, we can deduce the Granger causal relationship between soil moisture and temperature.

From Tables 6 and 7, we can see that at the significance level of 0.05 and 0.10, the past value of *MCSD30* helps predict *STSD30* and the past value of *STSD30* helps predict *MCSD30*. This suggests that the past value of soil moisture content is helpful in terms of predicting the present value of soil temperature at the depth of 30 cm, and the past value of soil temperature is helpful in terms of predicting the present value of soil moisture content at the depth of 30 cm.

From Tables 8 and 9, we can see that at the significance level of 0.01, 0.05, and 0.10, the past value of *MCSD90* helps predict *STSD90* and the past value of *STSD90* helps predict *MCSD90*. This suggests that the past value of soil moisture content is helpful in terms of predicting the present value of soil temperature at the depth of 90 cm, and the past value of soil temperature is helpful in terms of predicting the present value of soil moisture content at the depth of 90 cm.

## DISCUSSIONS

### Specific results

Firstly, the time lag of soil temperature's response to shock in soil moisture is about 25 h at 10 cm, 50 h at 30 cm, and 120 h at 90 cm, while the time lag of soil moisture's

**Table 7 Granger causality test determining whether *MCSD30* Granger causes *STSD30*.**

| Significance | *F*-statistic | Critical value | Conclusion |
|---|---|---|---|
| 0.01 | 4.2014 | 6.6415 | No |
| 0.05 | 4.2014 | 3.8439 | Yes |
| 0.10 | 4.2014 | 2.7068 | Yes |

Note:
The table displays the results of the Granger causality test of the depth of 30 cm. By comparing the value of *F*-Statistics and critical values at the significance levels of 0.01, 0.05, and 0.10, we can deduce the Granger causal relationship between soil moisture and temperature.

**Table 8 Granger causality test determining whether *STSD90* Granger causes *MCSD90*.**

| Significance | *F*-statistic | Critical value | Conclusion |
|---|---|---|---|
| 0.01 | 40.9051 | 2.6439 | Yes |
| 0.05 | 40.9051 | 2.0120 | Yes |
| 0.10 | 40.9051 | 1.7183 | Yes |

Note:
The table displays the results of the Granger causality test of the depth of 90 cm. By comparing the value of *F*-Statistics and critical values at the significance levels of 0.01, 0.05, and 0.10, we can deduce the Granger causal relationship between soil moisture and temperature.

**Table 9 Granger causality test determining whether *MCSD90* Granger causes *STSD90*.**

| Significance | *F*-statistic | Critical value | Conclusion |
|---|---|---|---|
| 0.01 | 10.5937 | 3.0220 | Yes |
| 0.05 | 10.5937 | 2.2164 | Yes |
| 0.10 | 10.5937 | 1.8488 | Yes |

Note:
The table displays the results of the Granger causality test of the depth of 90 cm. By comparing the value of *F*-Statistics and critical values at the significance levels of 0.01, 0.05, and 0.10, we can deduce the Granger causal relationship between soil moisture and temperature.

response to shock in soil temperature is for about 50 h at 10 cm and longer than 120 h at 30 and 90 cm.

Secondly, the intensity of soil temperature's impulse response to shock in soil moisture decreases from 0.2004 °C at the depth of 10 cm, to 0.0163 °C at the depth of 30 cm, and finally to 0.0035 °C at the depth of 90 cm. Similarly, the intensity of soil moisture's impulse response to soil temperature also decreases from 0.0638% at the depth of 10 cm, to 0.0163% at the depth of 30 cm, and finally to 0.0050% at the depth of 90 cm.

Thirdly, soil moisture is the Granger reason of soil temperature at the depth of 10, 30, and 90 cm. This means that the past value of soil moisture is helpful in terms of predicting the present value of soil temperature, and probably is the actual logical reason for changes in soil temperature. However, the causal relationship does not necessarily mean that there exists a direct connection between soil temperature and moisture, since a series of physical and biological processes may complicate the relations between soil temperature and moisture. While soil temperature does not help predict soil moisture at the depth of 10 cm, it helps predict soil moisture at the depth of 30 and 90 cm. Therefore, according to the basic principles of Granger causality test, when predicting the

dynamic variation of soil temperature and moisture at the depth of 30 and 90 cm, the other variable's past values should be taken into consideration, which means that a vector autoregression model may be a better choice. In comparison, at the depth of 10 cm, the past value of soil temperature is not helpful for predicting the present value of soil moisture. However, it may be more accurate to incorporate soil moisture as an exogenous variable when predicting soil temperature at 10 cm.

## Verification of the model

To begin with, the model we proposed can be supported and explained by established mechanistic models. In 1966, Philip proposed the concept of a complete SPAC, laying the theoretical foundation for modern farmland water research. Based on this, *Meng & Xia (2005)* established a dynamic coupling model by calibrating parameters that describes the hydrothermal conditions during crop growth and the law for crop transpiration. Through this coupling model, they hope to reveal the law for moisture and heat transfer of the SPAC. The SPAC is divided into three layers, namely, the atmosphere at a high altitude, the plant canopy which is simplified into one layer at the momentum transfer junction, and the soil layer. To be more specific, the top of the soil layer is set to be the soil surface, and the bottom of the soil layer is set to be at the groundwater level. According to the mathematical expression of total latent and sensible heat consumption of SPAC, the mathematical expression of plant transpiration latent and sensible heat consumption, the mathematical expression of soil evaporation latent and sensible heat consumption, and the model of soil moisture migration and heat transfer, the results obtained from data-driven vector autoregression models can be reasonably explained.

To begin with, the lagging effect of impulse response between soil temperature and moisture can be explained as follows. Moisture's lagging effect on temperature can be partly interpreted by the basic heat transfer equation:

$$C_v \frac{\partial T}{\partial t} = \frac{\partial}{\partial z}\left(K_h \frac{\partial T}{\partial z}\right) \tag{25}$$

where $T$ is soil temperature, $C_v$ is soil volumetric heat capacity, and $K_h$ is soil thermal conductivity. $C_v$ and $K_h$ are close correlated with soil moisture content $\theta$ and can be expressed in the single factor function form of $\theta$. From the equation above, we can deduce that when there is a fluctuation in soil moisture content, the first-order derivative of temperature $T$'s function on time $t$ will change. Thus, although the contemporary value of temperature will not change, it's successive value will change gradually in accordance with the change of the first-order derivative. Finally, at some point after the interference point, the deviation of temperature will reach its maximum, accounting for moisture's lagging effect on temperature. Temperature's lagging effect on moisture is to some extent related to the relationship between water vapor relative saturation and soil temperature.

$$h_2 = \exp\left(\frac{Mg\psi_2}{R(T_2 + 273.16)}\right) \tag{26}$$

where $h_2$ is the water vapor relative saturation of the air at the soil surface, $\psi_2$ represents water potential at the soil surface, $R$ is the universal gas constant, and $T_2$ denotes soil surface temperature. Though other possible reasons for the lagging effect may exist, one major process can be deduced from the expression above, where a fluctuation in soil temperature will first cause deviation in water vapor relative saturation, and then change in water vapor relative saturation will cause the corresponding variation in soil moisture through a series of complicated processes. In this way, a time lag exists.

The result that the time lag and intensity of the impulse response change with depth can be explained by several processes, some of which can be expressed as follows.

$$S_w = \left[ (4m - 1)/\mathrm{lr}(t) - (8m - 4)/\mathrm{lr}^2(t) \right] E_v(t) \tag{27}$$

$$E_v(t) = \int_0^{\mathrm{lr}(t)} S_w(z, t)\mathrm{d}z \tag{28}$$

Where $S_w$ is the water absorption intensity of roots, $m$ denotes the ratio of water absorption rate of the upper part to that of the lower part of roots, $\mathrm{lr}(t)$ represents root depth, and $E_v$ is crop transpiration rate. Equation (27) describes how $S_w$ is influenced by depth and time. Equation (28) describes the way in which plant transpiration is influenced by $S_w$ at different soil depths (*Lei, Yang & Xie, 1988*). Given that there exists a close relationship between soil temperature and plant transpiration latent and sensible heat consumption, it can be deduced that soil temperature varies with depth.

In addition to physical explanations, we used data from a different year and partly verified the universality of the model. We collected data on soil temperature and moisture content at the same test site from December 19th, 2016 to May 30th, 2017 at the depth of 10, 30, and 90 cm, during which the same irrigation method, that is, drip irrigation, has been adopted. Then, we used the same method stated above to build three vector autoregression models. Next, we built impulse response functions and conducted Granger causality tests. The results we obtained are as follows. First, the time lag of soil moisture's impulse response is 30, 70 h, and longer than 120 h, while the time lag of temperature's impulse response is about 60 h, longer than 120 h, and longer than 120 h at 10, 30, and 90 cm. Second, temperature's response intensity is 0.1989, 0.0157, and 0.0031 °C for 1% of variation in soil moisture, and moisture's response intensity is 0.0578%, 0.0169%, and 0.0057% for 1 °C of variation in soil temperature at 10, 30, and 90 cm. Third, soil moisture is helpful in terms of predicting soil temperature at the depth of 10, 30, and 90 cm. Besides, soil temperature is helpful in terms of predicting soil moisture at the depth of 10 and 30 cm but has no obvious relationship with soil moisture at 90 cm.

By comparing the results with those of 2017–2018, we can discover that the ratios for the intensity of soil temperature's impulse response to moisture among three depths during both periods approximate 200:16:3. The ratios for the intensity of moisture's impulse response to temperature during both periods is approximately 60:17:6. Moreover, the results of Granger causality tests are the same, indicating certain stability in terms of the physical and biological processes that involve the variation of soil temperature and

moisture in the same area. This demonstrates that for the same area the data-driven model we proposed has consistency and the results we obtained are not out of a sudden. Thus, the method of building data-driven vector autoregression models can be used to study the characteristics of soil heat-moisture coupling. Although the numerical results of this model will change with different soil properties in different areas, the method can be applied in the same way. The data and MATLAB code for training and testing the models have been included in the Supplementary Files for readers to verify our findings.

## CONCLUSIONS

We designed a purely data-driven stochastic model for analyzing moisture-heat coupling of a wheat-soil system, which consists of three vector autoregression models built at the depth of 10, 30, and 90 cm, impulse response functions, and Granger causality tests. For the empirical test of this method, we use the hourly data on soil moisture and soil temperature at the depth of 10, 30, and 90 cm obtained at Shangqiu Experiment Station. The following conclusions can be drawn from the models.

Firstly, the time lag of soil temperature's influence on soil moisture is for about 25 h at 10 cm, 50 h at 30 cm, and 120 h at 90 cm, while the time lag of soil moisture's influence on soil temperature is for about 50 h at 10 cm and longer than 120 h at 30 and 90 cm. Secondly, the intensity of soil temperature's impulse response to shock in soil moisture is 0.2004, 0.0163, and 0.0035 °C for 1% variation in soil moisture, respectively, at the depth of 10, 30, and 90 cm. Similarly, the intensity of soil moisture's impulse response is 0.0638%, 0.0163%, and 0.0050% for 1 °C of variation in soil temperature, respectively, at the depth of 10, 30, and 90 cm. Thirdly, soil moisture is helpful in terms of predicting soil temperature at the depth of 10, 30, and 90 cm. While soil temperature helps predict soil moisture at the depth of 10 and 30 cm, it has no obvious correlation with soil moisture at 90 cm.

Although the proposed method for analyzing moisture-heat coupling in a wheat-soil system has some advantages such as data-driven and easy to achieve, there are limitations. One is that the vector autoregression model is based on the linear hypothesis, so other nonlinear data-driven models should be further studied. Also, since the soil of our experiment site is loam, we ought to further explore whether such a method can be applied to other kinds of soil using data from different experiment sites. We are currently working on this.

### Funding

This work was supported by the National Key Research and Development Program of China (2017YFD0301102-5), the Philosophy and Social Science Planning Project of Henan Province, China (2018BZH005), the Science and Technology Research Project of Henan Province (182102110029), and the Doctoral Research Fund of North China University of Water Resources and Electric Power, China (4001/40534). There was no additional external funding received for this study. The funders had no role in study design, data collection and analysis, decision to publish, or preparation of the manuscript.

## Grant Disclosures

The following grant information was disclosed by the authors:
National Key Research and Development Program of China: 2017YFD0301102-5.
Philosophy and Social Science Planning Project of Henan Province, China: 2018BZH005.
Science and Technology Research Project of Henan Province: 182102110029.
Doctoral Research Fund of North China University of Water Resources and Electric Power, China: 4001/40534.

## Competing Interests

There are no competing interests.

## Author Contributions

- Xiaohang Feng conceived and designed the experiments, performed the experiments, analyzed the data, contributed reagents/materials/analysis tools, prepared figures and/or tables, authored or reviewed drafts of the paper, approved the final draft, xiaohang Feng refines the essay.
- Xia Zhang conceived and designed the experiments, performed the experiments, approved the final draft.
- Zhenqi Feng conceived and designed the experiments, approved the final draft.
- Yichang Wei conceived and designed the experiments, performed the experiments, approved the final draft.

## Data Availability

The raw data are available in the Supplemental Files. The raw data show soil moisture and soil temperature obtained at our Shanqiu test site from 2016 to 2018. These data are used to build vector auto-regression models at the depth of 10, 30, and 90 cm and analyze heat-moisture coupling of a wheat-soil system. The data can be tested using MATLAB (https://www.mathworks.com/downloads/). The MATLAB code for testing the results are available in the File S3. All of the materials described above are also available at https://github.com/florafeng2016/VAR-model-heat-moisture-coupling.

## Supplemental Information

Supplemental information for this article can be found online at http://dx.doi.org/10.7717/peerj.7101#supplemental-information.

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
