# Peer review of "Analyzing moisture-heat coupling in a wheat-soil system using data-driven vector autoregression model"

_PeerJ, doi:10.7717/peerj.7101_

## Round 0.1 · original submission · Major Revisions

The paper was evaluated by 4 reviewers and all suggested major revision. The major issue is that the authors have not linked the data driven approach to soil physical models. There are many models based on soil physics that deal with the topic of moisture and temperature coupling. Linking data-driven to mechanistic approach would be valuable.

Reviewer 1 ·

Basic reporting

No comments.

Experimental design

No comments.

Validity of the findings

No comments.

Additional comments

This manuscript analyzes moisture-heat coupling in a wheat-soil system using data-driven vector auto-regression model. The manuscript is not well written. There are some comments as below,

In Introduction part, please add some most recent references regarding the heat and moisture transfer process and summarize their major findings and unsolved critical problems.

Discussion part should be revised thoroughly, the reviewer could not find any deep analyses and discussion in terms of the results obtained in the previous sections.

Conclusion part is redundant and needs revision. Only major findings resulted from the study should be included in the conclusion.

English editing by a native speaker is highly recommended.

Reviewer 2 ·

Basic reporting

This article proposed a new model to coupling the water content and temperature in wheat soil. It is significant to develop this data-driven model to research the interaction of soil moisture and temperature in field.

But there are also some questions and unclear about this article when I read this paper. Some doubts have been marked in the text. Others have been written in the following parts (Experimental design and validity of the findings).

Experimental design

1. You also said that you did not use other models to verify this new model at the end of the article. How to prove the results you obtained were reliable?

2. In the part “Field experiment on soil moisture and temperature”, the experiment was very clear. In the article you referred the drip irrigation and flood irrigation, and analyzed the influence of two irrigation systems on the coupling of soil water content and temperature. However, you haven’t mentioned the irrigation in the experimental design. Were there two irrigation systems in this experiment? And how-to analysis the different effect of drip and flood irrigation on soil moisture and heat?

Validity of the findings

1. In this article, the time lag and response intensity are the most important results to indicate the interaction of soil water content and heat at different soil layer. Therefore, in the abstract you should explain the meaning of these results, and how to indicate the effect of soil moisture and heat. (Line18-30)

2. During designed the stochastic data-driven model, I am not very clear the process of developing, and how to calculate the time lag and response intensity using the soil water content and temperature data. Maybe I was not good at math and modeling. So, this part could be more detailed. (Line 81)

3. In the figures, you should be better to specify the horizontal and vertical coordinates, despite you have explain the meaning in text. (Figure 1-3)

4. You must be labeling the units of physical quantities, soil moisture and temperature.

Additional comments

I think it is a good idea to design this new model to analyze the coupling of soil moisture and heat. There are also lack of some data of different year and other depth. We should to verify the new model at different soil texture, depth and irrigation system. In addition, we should utilize other model which have been used widely to test the accuracy of new model.

Annotated reviews are not available for download in order to protect the identity of reviewers who chose to remain anonymous.

Reviewer 3 ·

Basic reporting

Theoretically, soil moisture content dependent soil thermal diffusivity affects significantly thermal signals propagation from ground surface, thus soil temperature in soil profile and soil moisture content of corresponding soil layer related closely each other. There are many studies focused on unknown soil temperature prediction from known soil temperature, soil water content and other soil properties, such as Hu and Isram, 1995; Hirota et al., 2002; Holmes et al., 2008; Wang, 2012; Huang et al., 2014. Also many previous studies devoted to soil water content inversion from temperature measurements, like Steele-Dunne et al., 2010; Krzeminska et al., 2012; Bechkit et al., 2014; Halloran et al., 2016; Dong et al., 2015; Bristow, 1998; Ren et al.,2000; Liu,2011; Striegl and Loheide, 2012; Sayde et al., 2014. However due to the relationship between them involved many complicated factors, like soil properties, soil heat and water coupling transport, canopy and so on. A direct relationship between them is difficult to be established. Thus the results like P344-346 and 382-390 stated wrongly.

This manuscript investigates the relationship between present value and past value of soil temperature and soil moisture contents (different depth of soil profile). It provides the opportunity to understand the lagged effect of temperature on soil moisture content, or vice versa. Some results of soil profile during wheat greening period are obtained. In this regard, I like the contribution. However, the paper fails to verify these regression equations, and provides no general relationship between present value and past value of temperature and moisture content. Because the complexity of both crop factor and heat and water transport. Maybe the regression equation is rather empirical, thus all the conclusions are lack of universality. Like statement in the introduction section that “some soil moisture-heat coupling simulation, such as Philip and De Vries (1957), Nassar and Horton (1989), Shang et al. (1997), Bittelli et al. (2008)”. These studies estimates soil water and temperature parameters using numerical simulation of water-heat or water-heat-salt transport equation, which based on soil physical process and thus are a general method. Thus I really worried about the contribution of this paper.

Several issues in the format. (1) There should a space between number and units like h and cm but no space between number and oC. (2) some sentences are lack of a punctuation for broken sentences such as Lines 401. (3) The figures have no axis titles and units.

Finally, the manuscript is poorly organized and expressed. Some sentences in the Discussion and Conclusion section duplicate the information from the Results section. Many Grammatical mistake occurred, for example,

Line 51. What is affected closed soil column.

Lines 52-53. This statement is difficult to understand.

Line 127. Increase the likelihood of what? Fitting models? Doing so may result in over-fitting? Unclear statement

Lines 131-133. What is estimated model and estimated regression model.

Line 155. The impulse response can be built. Improper statement.

Lines 160-161. The explanatory ability of regression. What is meaning?

Line 186. The time length during which there is obvious response of soil moisture. What is meaning?

Line 400. For the empirical test of this method. What is meaning?

Experimental design

no comment

Validity of the findings

The authors also spent effort on providing suggestions on irrigation using the analyzing results from regression equations. This simple function is far from enough to guide irrigation as it is affected by many factors such as soil and crop properties.

Additional comments

This paper maybe better if focused on analyzing the lagging effect between soil moisture and temperature using mentioned method, and also verify its universality and potential importance using more data of different conditions. The author also should improve the english writting to a greater extent.

Reviewer 4 ·

Basic reporting

Clear and concise. Sufficient literature cited. Professional figs, tables.


Raw data and codes are not shared.

Experimental design

The experiment design is not very sound because the authors have not discussed the heterogeneity of the soil horizontally, and particularly vertically.

Validity of the findings

The authors should mention/discuss how the results discovered here may vary with soil texture and climate conditions.

Additional comments

In this study, the authors used a data-driven vector auto-regression model to investigate the moisture-heat coupling in a wheat-soil system. They found that the time lags and response intensity for moisture and temperature responses were different at different depths. Based on the results, they suggested that surface flooding irrigation is appropriate and the vector auto-regression model can help predict soil moisture response at 10 and 30 cm.

In general, the manuscript is well written. However, some minor grammar mistakes need attention. For example, the last sentence of the abstract and line 146 (tow variables?). I suggest the authors recheck the grammar throughout the paper and improve the readability of the manuscript, particularly the method section.

In addition, I have a few comments on the manuscript.

Lines 183: What are the six impulse response functions? Temperature as shock and moisture as response vs. moisture as shock and temperature as a response? Are the shocks all positive or negative?

Lines 192: why 30 cm? Please describe the irrigation systems first, particularly the depths of the drip irrigator. Did you only have drip irrigation at 30 cm?

Lines 218 to 225: Provide more details on the principles of the sensors. What type of sensors did you use to measure soil moisture and temperature? How did you know they are “highly accurate” and the measurements are free of instrumental drifts? What is the evidence? The measurement accuracy of 0.5 degrees is not very accurate. How about the accuracy of soil moisture?

Lines 246. Please plot the time series after you removed the seasonality.

Figures 3 and Table 3: Please define or plot the shock signals for soil moisture and temperature at different depths for different scenarios. When you say one unit positive shock, what is the unit? What are the units of responses in Table 3?

The results presented here are based on measurements of soil moisture and temperature at one location. How did you deal with the spatial variability of the soils horizontally and vertically? Do you have any physical and chemical analysis data on the soil?

The authors should mention/discuss how the results discovered here may vary with soil texture and climate conditions.

Lines 366 to 372: What are the physical implications of these lags and intensity? Please try your best to explain these results using the previous studies you cited in the introduction such as Liu et al. (2005).

Lines 373-381: This is a key section of the manuscript. Please provide more details on the management issues related to the coupling of soil moisture and temperature in irrigation and discuss more previous studies related to this issue. Why do you want to avoid/trade off the influence of soil temperature on soil moisture when using surface flood irrigation? Would you see the same effect if the soil were within a warming or cooling phase at the time of irrigation? For example, an upward temperature gradient occurs in the winter before the sunrise will affect the water movement in the soil but this will be different after the sunset.

Lines 382-390: How did your results contribute to the current knowledge of coupled heat and water movement in soils as discussed by the previous studies and cited in the introduction? Are you suggesting that it is possible to include the hysteresis effects for an updated coupled heat-water-vapor movement model in the soil as suggested by researchers such as Liu et al. (2005) and Nassar and Horton (1989)?

Since PeerJ is an open access journal, the authors should refer to the online software packages that can be used to conduct this type of analysis or upload a simplified version of the codes they used to generate the results. In this way, more researchers from soil and hydrological sciences can easily include the coupling of soil moisture and temperature for their research. In addition, this will help get the paper noticed by the peers and increase the citation of the work.

---

## Round 0.2 · Minor Revisions

The reviewers suggested some remaining clarification of the manuscript

Reviewer 2 ·

Basic reporting

No comments.

Experimental design

No comments.

Validity of the findings

No comments.

Additional comments

In the revised manuscript,the authors linked this data-driven model to the physical model, SPAC Continuum, and explained the results obtained from the vector auto-regression model using four equations in the SPAC Continuum. They proved that the results obtained from the vector autoregression model are reasonable with sound physical foundations, especially considering the processes of soil evaporation and plant transpiration.
This aim of the article is to introduce an acceptable method to research the relationship of soil water and heat. Despite the authors has revised the article based on the comments of reviewers, the article is a bit of complicated and verbose. Especially the abstract part, the results is not refined enough. They added much new information to explain the model and others, but this make the process more complicated. So I think it is better to rearrange the structure of the article, and this article can be accepted.
The English writing has improved a lot.

Reviewer 3 ·

Basic reporting

Part of the expression is a lack of linguistic rigor: your study focused on the relationship between the present value of two variables and each other’s past value rather than soil temperature and soil moisture. Thus the expression, such as line 363 and 400, it should be the past value of soil temperature is not help for predicting present value of soil moisture. From physical perspective, soil temperature and soil moisture is related closely each other.

Experimental design

no comment

Validity of the findings

The author still failed to verify the universality of vector autoregression model and conclusion. For the same experiment site but different years, the author obtained different results like lines 395-398 (results of 2016-2017) and lines 411-413 (results of 2017-2018). Thus the author should be cautious about the model’s development, maybe it is very empirical.

Additional comments

In Introduction part, please logically simplify this part about the previous researches and relate it with your study, rather than just a long list.

Double check some problems like:
Line 235, there still lack space between number and unit of cm.
Line 245, volumetric water content unit should be m3 m-3 rather than %
Line 467, unifying the unit of time with before (hours and h).
Lines 367-368, the depth should be 30 cm rather than 10 cm

---

## Round 0.3 · Minor Revisions

The authors have revised the manuscript accordingly. The English still needs polishing.

e.g.
L. 26, soil moisture Granger causes soil temperature at 10.. doesn;t make sense
L. 28 SPAC is soil plant atmosphere Continuum!
L. 30-31 the dynamic of what?
L. 36 They are not conceptual models, but mechanistic models. What's wrong with that?
L.36-69 This needs to be rewritten as it is very dense to read. Please briefly summarise existing studies and point out the need for empirical models.
L.59 " result can only be applied to sandy soils" The same with this paper, it can only be applied to soils studied in this paper
etc..
etc..

I suggest the authors need to seek help from a native English speaker to make the manuscript more readable by international audience.

---

## Round 0.4 · Minor Revisions

The paper still has lots of snags. Please see my corrections.

---

## Round 0.5 · Minor Revisions

While the paper has been revised. there is still a few grammatical errors, which I indicated in the attachment. Please revise this. Note : there is no such thing as "powder" in soil. It should be silt!

The Section Editors for the journal have looked at your submission and as a result we request that your code/test data be provided before we can Accept the submission. In order for validation of the model to move forward the test data must be provided so that it can be a vehicle for others to test it. The manuscript is in generally good order, except that without a way for others to test the findings there is no substance to it.

---

## Round 0.6 · Minor Revisions

One of the Section Editors wrote:

"In order for validation of the model to move forward the test data would need to be provided as it would be a vehicle for others to test it. The manuscript was in generally good order, except there is no meat to it without a way to test the findings. Codes and data are always a good thing"

Therefore we ask the authors to provide the computer codes as supplementary materials so readers can use them.

---

## Round 0.7 · accepted · Accept

The authors have provided data and computer codes that facilitate replication of this research.